# External Root Resorption Management of an Avulsed and Reimplanted Central Incisor: A Case Report

**DOI:** 10.3390/dj9060072

**Published:** 2021-06-16

**Authors:** Gianni Di Giorgio, Alessandro Salucci, Gian Luca Sfasciotti, Flavia Iaculli, Maurizio Bossù

**Affiliations:** 1Department of Oral and Maxillofacial Science, “Sapienza” University of Rome, Via Caserta, 6, 00161 Rome, Italy; gianni.digiorgio@uniroma1.it (G.D.G.); alessandro.salucci@uniroma1.it (A.S.); gianluca.sfasciotti@uniroma1.it (G.L.S.); maurizio.bossu@uniroma1.it (M.B.); 2Department of Neuroscience and Reproductive and Odontostomatological Sciences, University of Naples “Federico II”, Via Pansini, 5, 80131 Naples, Italy

**Keywords:** dental trauma, root resorption, tooth avulsion, tooth reimplantation

## Abstract

**Background**: Avulsion and reimplantation of permanent teeth represent a major challenge in terms of treatment and long-term prognosis. The present study reported clinical management of external root resorption of an avulsed and reimplanted maxillary central incisor. **Case report**: A 9-year-old boy reported an uncomplicated crown fracture and avulsion of tooth 11 and complicated crown fracture of tooth 21 due to trauma. Reimplantation of element 11 was obtained within 30 min post-trauma and 3 days after both elements were diagnosed with necrotic pulp. In addition, tooth 11 showed early external root resorption. Both elements underwent endodontic treatment and root closure with apical plug using calcium-silicate-based cement. At 6-month follow-up root resorption appeared to be arrested. Twenty-four months after trauma the clinical results were stable, although signs and symptoms of ankylosis were observed. **Conclusions**: An immediate endodontic approach and use of calcium-silicate-based cement seemed to contrast the progression of root resorption of an avulsed and reimplanted central incisor after 24 months of follow-up.

## 1. Introduction

Trauma involving the dento-alveolar region represents a major emergency in dentistry, with a frequent occurrence in children and young adults. Among others, one of the most serious issues is avulsion of permanent teeth, which constitutes 0.5–16% of all dental injures [1].

Replantation of avulsed teeth represents nowadays the treatment of choice; however, some factors should be taken into account that might interfere with the prognosis and long-term survival of the dental element, such as maturity of the root (open or closed apex), condition of periodontal ligament (PDL) and extra-oral storage [1,2]. The condition of PDL cells is mainly related to the extra-oral time and medium storage; indeed, International Association of Dental Traumatology Guidelines [1] highlighted that, after 30 min of extra-oral dry condition, loss or damage of PDL cells should be evaluated before reimplantation. In addition, the maturity of the root determinates the treatment plan, since in case of immature permanent tooth the healing potential of the root may be applied in terms of revascularization after replantation, and endodontic treatment should be delayed [3]. On the other hand, reimplanted dental elements with closed apex should be endodontically treated within 2–3 weeks after trauma, considering a potential contamination of the necrotic pulp after avulsion [1].

One of the main drawbacks of reimplantation after avulsion is represented by external root resorption [4], which can be further divided into surface, inflammatory, or replacement (ankylosis) root resorption [5]. The latter demonstrated an incidence of 51% [5] and it is strictly related to the extra-oral dry time and the extent of PDL damage, mostly causing tooth fusion to alveolar bone [6]. Early diagnosis is the most critical factor to help rapid decision on the best treatment approach to be initiated and to mitigate the consequences of the continuing progress of root resorption [7,8]. Indeed, treating and arresting root resorption represents a major challenging in endodontic procedures and plays a pivotal role in the long-term survival of reimplanted dental elements. Use of calcium-silicate-based cements in the therapy of external root resorption was suggested to improve the treatment outcomes [7,9]. Bioactive materials induce a high pH which would impair the osteoclastic action resulting in the arrest of resorptive process [10] and increasing the success over time.

Therefore, the aim of the present study was to report a case of external root resorption of an avulsed and reimplanted maxillary central incisor, and to describe its clinical management.

## 2. Case Report

The present case report was reported following Preferred Reporting Items for Case reports in Endodontics (PRICE) guidelines [11] (Table A1).

A 9-year-old boy was referred to the Unit of Pediatric Dentistry, Policlinico Umberto I—Sapienza University of Rome reporting a traumatic accident in school 3 days before, which caused uncomplicated crown fracture and avulsion of tooth 11 and complicated crown fracture of tooth 21. Tooth 11 was reimplanted in a private practice 30 min post- trauma and then stabilized using orthodontic brackets and a passive flexible splint. The medical history was not relevant.

Clinical examination revealed lesions of extra- and intra-oral soft tissues. Pulp sensibility tests were negative to cold, percussion, and palpation for both elements; periapical radiograph showed a slight area of radiolucency involving the external root walls of tooth 11, consistent with an early external root resorption. Moreover, periapical radiolucency involving tooth 21 was noticed (Figure 1). A diagnosis of necrotic pulp was made for both elements. Treatment procedures were explained to patient’s parent and a signed informed consent was obtained.

A week after, both elements underwent root canal treatment. After administration of local anesthesia (epinephrine-free mepivacaine) at pericoronal gum, dental elements were isolated by rubber dam and pulp chambers were opened using high-speed round bur under abundant irrigation. Working length was determined by means of wide manual endodontic file (#80) (Figure 2). Each tooth was irrigated with 5 mL of 5.25% sodium hypochlorite (NaOCl) using a 5 mL plastic syringe supporting a 30-G irrigation needle (Ultradent Products Inc., South Jordan, UT, USA), then saline solution was flushed and removed by aspiration. Finally, irrigation with 5 mL of 17% ethylenediaminetetraacetic acid (EDTA) was performed. Tooth 21 underwent also manual shaping of the root canal. After drying with paper points, an intracanal dressing material made of calcium hydroxide (Stomidros—Stomygen, COSWELL SPA, Funo, Italy) was placed. Then, temporary polymer reinforced zinc oxide–eugenol (IRM, Dentsply International Inc., Charlotte, NC, USA) restorations were placed.

Three weeks after the first appointment, periapical radiographs were obtained (Figure 3) and both elements underwent rubber dam isolation, reopening and endodontic disinfection as described before with 5.25% NaOCl and 17% EDTA.

After drying with paper points, apical plugs were obtained by the placement of calcium-silicate-based cement (ProRoot MTA, Dentsply-Sirona, Charlotte, NC, USA) (Figure 4). Moistened paper points were placed over the cement and teeth were temporally re-stored by polymer reinforced zinc oxide–eugenol cement (IRM).

One week after, both elements were isolated by rubber dam and re-opened. Paper points were removed and, once cement hardening was clinically confirmed by a dental probe, root canals were backfilled with gutta-percha (Figure 5) and dental elements temporally restored.

The following week (6 weeks post-trauma), splinting was removed and absence of tooth mobility was clinically observed. Then, crowns of both dental elements were definitively restored by direct composite resin esthetic reconstructions (Figure 6).

At 6-month follow-up, a periapical radiograph reveled stability of the applied materials and absence of periapical lesions of teeth 11 and 21; moreover, external root resorption involving tooth 11 appeared to be arrested (Figure 7).

At the last follow-up visit 24 months post-trauma, absence of tooth mobility as well as signs and symptoms of inflammation were clinically observed. However, infraocclusion of tooth 11 was observed. The radiographical evaluation revealed a slight area of radiolucency and absence of periodontal ligament of tooth 11 (Figure 8). These findings, in addition to infraocclusion and absence of mobility, might suggest a diagnosis of anchylosis.

## 3. Discussion

Traumatic dental injures involve more than one billion people worldwide and represent a major issue from a medical as well as economical point of view [12]. Thus, the correct management of dental trauma would not only reestablish the subjects’ health but also decrease the long-term side effects and improve the prognosis.

Reimplantation of avulsed permanent teeth is considered the treatment of choice mainly in young patients, whose development of the maxillofacial region is not completed. However, root resorption is a well-established consequence of tooth reimplantation [5] and should be correctly treated to guarantee a better prognosis over time [1]. Mazur et al. [13] reported a high heterogeneity in root resorption pattern of reimplanted central incisors due to several factors, probably related to extra-oral dry time and storage, and concluded that the poor prognosis often occurred when teeth are replanted not in accordance with current guidelines for the management of dental avulsion [1]. In the present case report, to contrast the progression of external root resorption, reimplanted tooth underwent root canal treatment just 1 week after the occurrence of trauma. This prompt approach taken into account that the rate of root resorption is related to age and was demonstrated to be significantly higher in subjects of 8–16 years old at the time of avulsion [14]. Moreover, the apical plug was obtained using a bioactive calcium-silicate-based cement that promotes the deposition of a mineralized tissue that would contrast the progression of resorption process [15]. Bioactive cements, among others MTA, demonstrated better performances than calcium hydroxide when applied to treat external root resorption of reimplanted teeth [16]. Accordingly, in the present case, although the radiographical evaluation 24 months post-trauma revealed a slight area of radiolucency involving the root of tooth 11, the resorption rate seemed to be less than that of the early stages. However, signs and symptoms of ankylosis related root resorption were appreciated.

Risk of ankylosis is a well-documented complication after tooth avulsion and reimplantation [17] and it is strictly associated to extra-oral dry time and reimplantation management. Ankylosis might be highly avoided in case of reimplanted immature teeth with dry time longer than 60 min [18]. However, once ankylosis is present, decoronation may be necessary according to the evidence of esthetically unacceptable infra-occlusion that cannot be corrected by restorative treatment [1,19]. It should also be stressed that immature permanent teeth demonstrated lower risk of ankylosis [17] and reimplementation is considered the treatment of choice. In addition, even though exposed to a major risk of ankylosis, mature permanent teeth should be also reimplanted in young patients, considering the not completed development of the maxillofacial area.

Although in the present case report the avulsed and reimplanted immature permanent tooth underwent apexification, revitalization of necrotic open apex teeth might be considered a valid alternative treatment to the apical plug with MTA [2]. Regeneration procedures have been recently proposed also in mature permanent teeth [20], but they should be further confirmed in case of avulsion and reimplantation. In addition, a regenerative endodontic approach has been very recently suggested as a treatment strategy to arrest progressive development of root resorption of young permanent teeth, with the aim to promote new periodontal ligament attachment to ankylosed teeth [4]. These preliminary results should be supported by further evidence to make endodontic regeneration a reliable alternative in the management of traumatized teeth, improving the prognosis over time and decreasing the occurrence of side effects and need of early surgical intervention.

## 4. Conclusions

Within the limitation of the present case report, an immediate endodontic approach and use of calcium-silicate-based cement would reduce the progression of root resorption of an avulsed and reimplanted central incisor after 24 months of follow-up.

## Figures and Tables

**Figure 1 dentistry-09-00072-f001:**
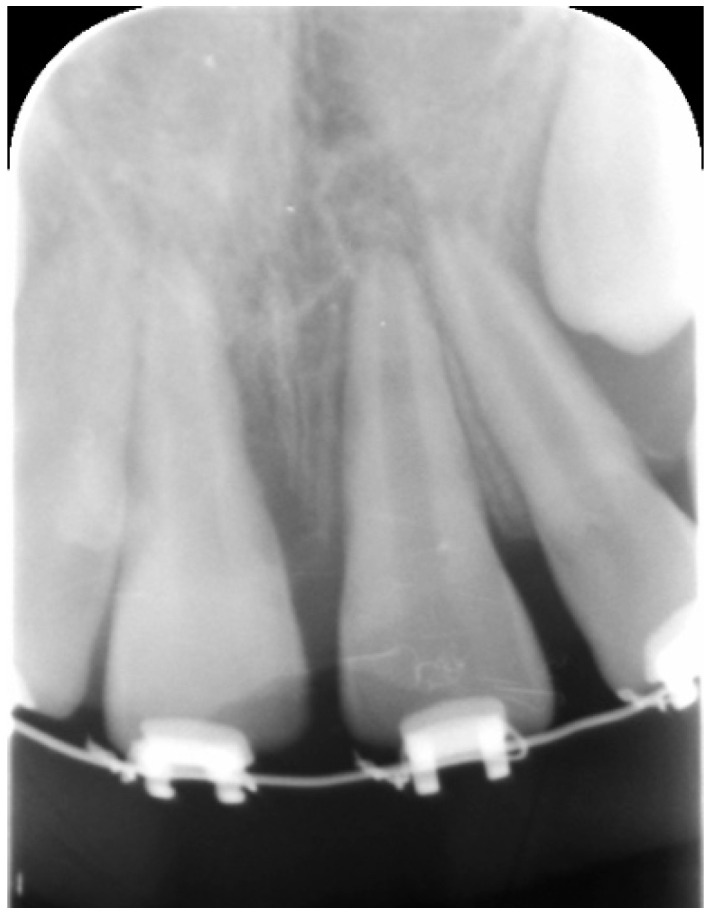
Periapical radiograph showing external root resorption involving external root walls of tooth 11 and periapical radiolucency involving tooth 21.

**Figure 2 dentistry-09-00072-f002:**
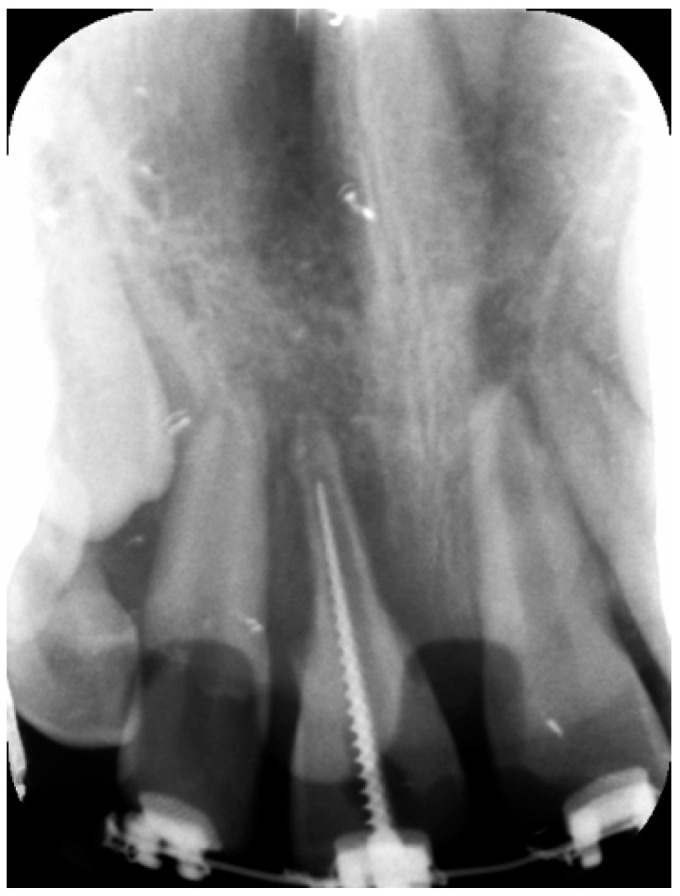
Periapical radiograph during root canal treatment demonstrating an aggressive external root resorption of tooth 11.

**Figure 3 dentistry-09-00072-f003:**
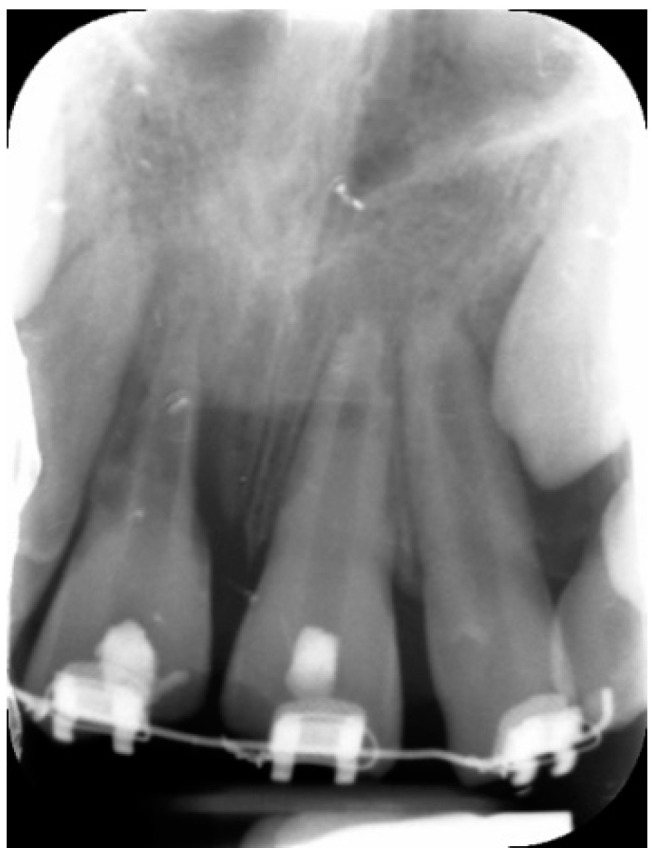
Periapical radiograph 3 weeks after the first endodontic approach, showing a decrease in progression of external root resorption of tooth 11.

**Figure 4 dentistry-09-00072-f004:**
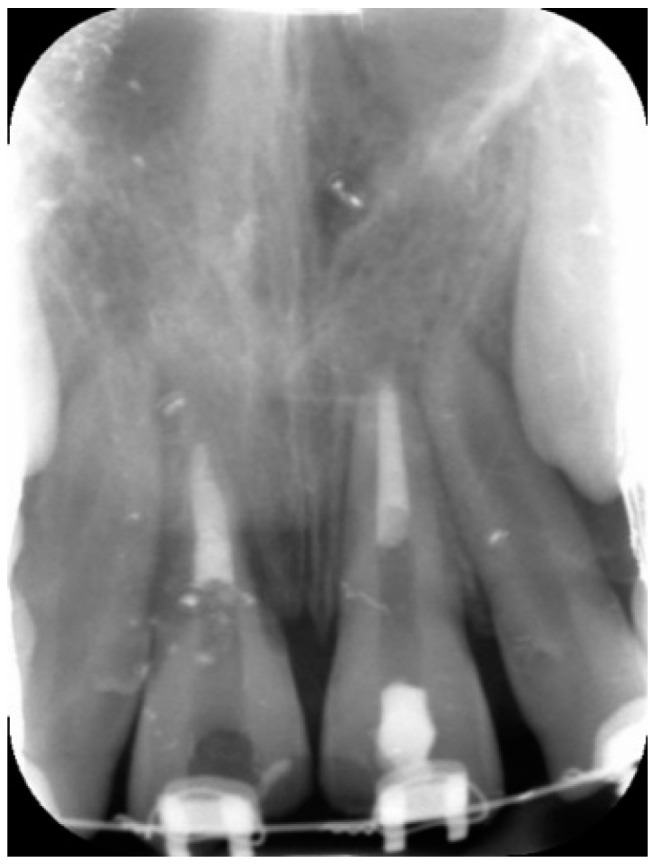
Periapical radiograph demonstrating an apical plug of both dental elements (11 and 21) obtained with MTA.

**Figure 5 dentistry-09-00072-f005:**
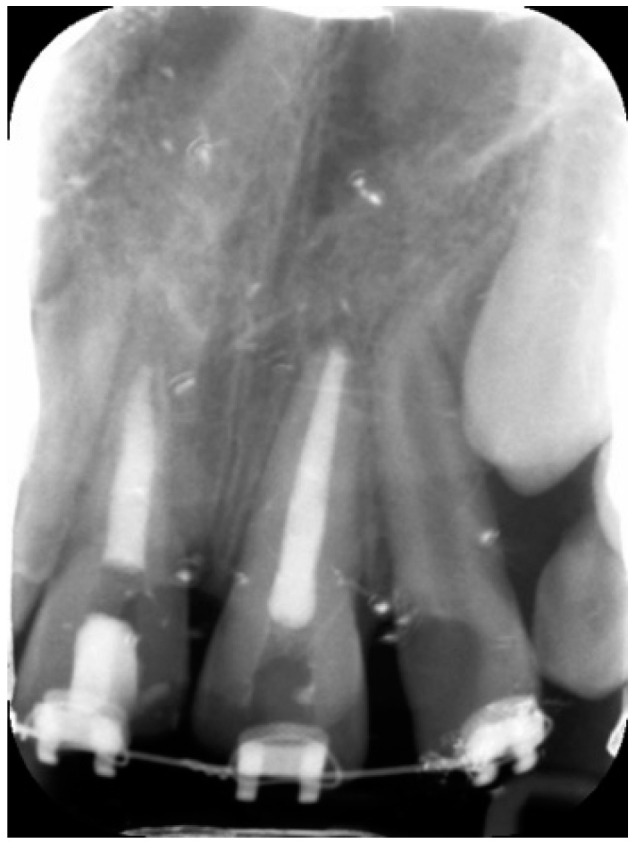
Periapical radiograph demonstrating the conclusion of the endodontic treatment of teeth 11 and 21, respectively.

**Figure 6 dentistry-09-00072-f006:**
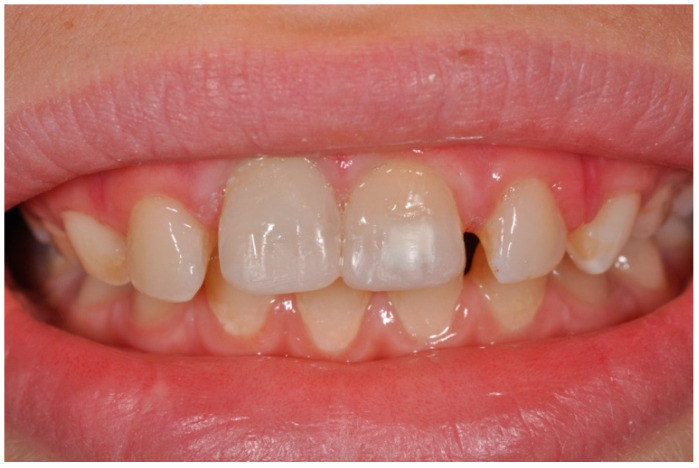
Clinical image showing the definitive restorations of 11 and 21 dental crowns.

**Figure 7 dentistry-09-00072-f007:**
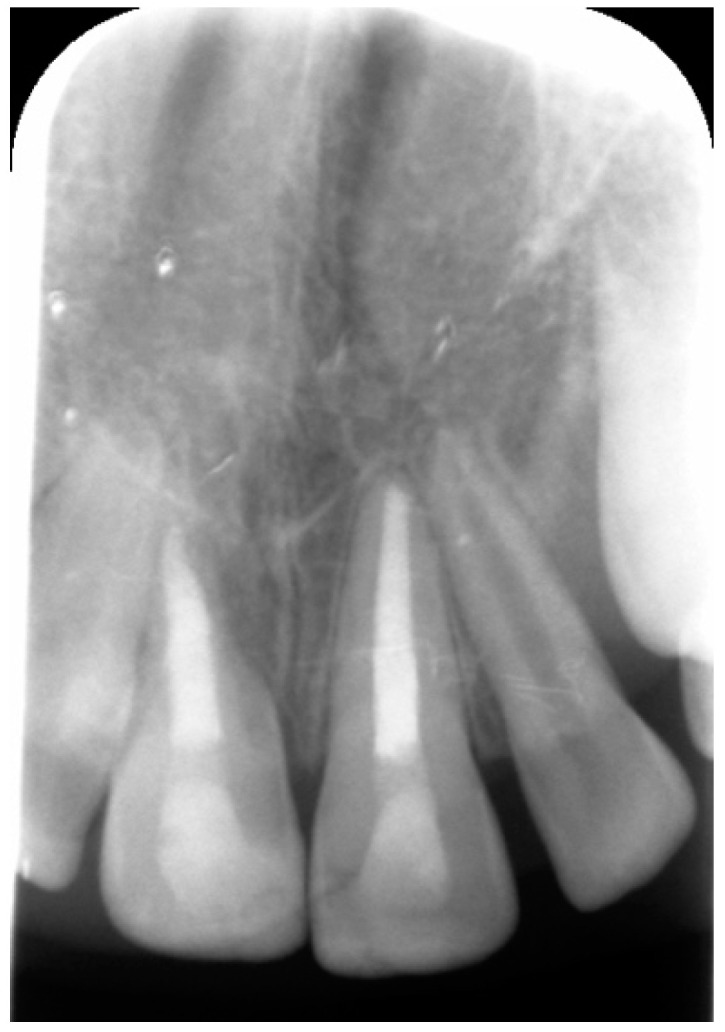
Radiographical follow-up 6 months after treatment.

**Figure 8 dentistry-09-00072-f008:**
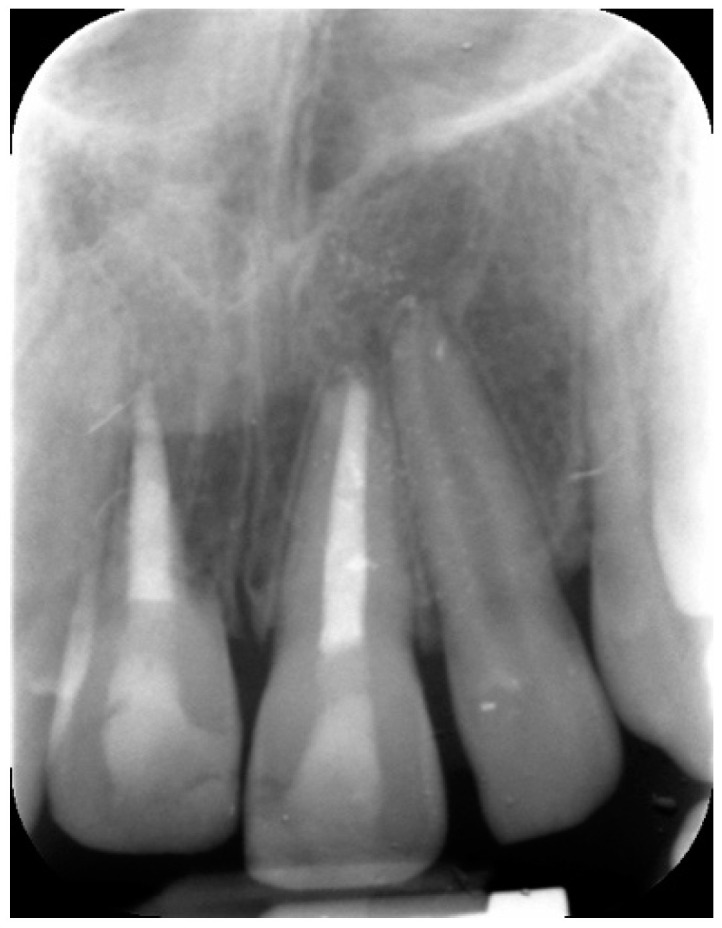
Radiographical follow-up 24 months post-trauma.

## Data Availability

Data is available on reasonable request.

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
