# Peer review of "External Root Resorption Management of an Avulsed and Reimplanted Central Incisor: A Case Report"

_dentistry, 2021, doi:10.3390/dj9060072_

Round 1
Reviewer 1 Report
This manuscript describes the treatment of external root resorption of an avulsed and reimplanted tooth. Dental trauma diagnosis and treatment is a very important topic and this is under the scope of the journal. The case report is interesting and the manuscript is mostly well structured.
However, there are some issues that need to be solved before publication:
1-At introduction, authors must stress that an early diagnosis is the most critical factor to help rapid decision on the best treatment approach to be initiated, to mitigate the consequences of the continuing progress of root resorption (DOI 10.14744/eej.2018.33043). Also, some information concerning the new opportunities to improve the treatment outcome with the use of bioactive hydraulic calcium silicate cements needs to be presented;
2-Table 1 should be presented as a supplemental table, there’s no reason to be part of the main manuscript;
3-P4L67 – There is an error in this description, because it is clear from the radiograph that tooth 21 is associated with a periapical radiolucency;
4-Legend of Figure 1 needs to give some explanation about tooth 21;
5-Irrigation protocol needs to be better defined: did authors neutralize the NaOCl before irrigation with chlorhexidine? Which was the amount of irrigant used and delivery system?;
6-P7L153- Typo error, reimplantation;
7-At discussion section, authors most address the potential discoloration problems caused by the use of MTA and present alternatives to minimize this problem with other hydraulic calcium silicate cements.
8-At conclusions, authors need to choose a better word then “contrast” to express their view.
Author Response
This manuscript describes the treatment of external root resorption of an avulsed and reimplanted tooth. Dental trauma diagnosis and treatment is a very important topic and this is under the scope of the journal. The case report is interesting and the manuscript is mostly well structured.
However, there are some issues that need to be solved before publication:
The Authors are thankful for the Reviewer’s considerations and the manuscript has been improved as suggested by the Reviewer.
1-At introduction, authors must stress that an early diagnosis is the most critical factor to help rapid decision on the best treatment approach to be initiated, to mitigate the consequences of the continuing progress of root resorption (DOI 10.14744/eej.2018.33043). Also, some information concerning the new opportunities to improve the treatment outcome with the use of bioactive hydraulic calcium silicate cements needs to be presented;
Introduction section has been improved as recommended and the suggested paper has been quoted within the manuscript.
2-Table 1 should be presented as a supplemental table, there’s no reason to be part of the main manuscript;
Table 1 has been removed from the text and presented as supplementary material and quoted within the manuscript as Table S1.
3-P4L67 – There is an error in this description, because it is clear from the radiograph that tooth 21 is associated with a periapical radiolucency;
The description of periapical radiolucency involving tooth 2.1. has been edited as recommended.
4-Legend of Figure 1 needs to give some explanation about tooth 21;
Figure 1 caption has been amended according to the text, as mentioned before.
5-Irrigation protocol needs to be better defined: did authors neutralize the NaOCl before irrigation with chlorhexidine? Which was the amount of irrigant used and delivery system?;
Irrigation protocol has been better described and provided within the Case Report section according to the Reviewer’s suggestions.
6-P7L153- Typo error, reimplantation;
Error has been checked and edited.
7-At discussion section, authors most address the potential discoloration problems caused by the use of MTA and present alternatives to minimize this problem with other hydraulic calcium silicate cements.
The Authors agree with the Reviewer’s concern about the problem of discoloration caused by MTA. However, this issue is mostly reported when MTA is applied in vital pulp therapy (direct/indirect capping and pulpotomy) or in perforation repair, since, in these cases, MTA is used in the area of the pulp chamber and may result in further discoloration of the tooth crown. On the other hand, when MTA is applied in endodontic surgery or apexification procedures, as in the present case report, the MTA-related discoloration problem could be neglected. Due to these considerations the aforementioned topic has not been developed within the Discussion section.
8-At conclusions, authors need to choose a better word then “contrast” to express their view.
The word “contrast” has been replaced by “reduce” according to the Reviewer’s suggestion.
Reviewer 2 Report
This case reports demonstrates a treatment strategy regarding a necrotic immature permanent tooth with delayed replantation. The topic is of clinical significance in the field of pediatric dentistry. However, there remained some concerns.
1. I recommend to revise tooth numbering system for readability. Please consider to provide FDI-two digit tooth numbering system or the full name of the teeth.
2. Please provide a PRICE 2020 checklist (table 1) as supplementary files, instead of table.
3. In the Page 6, Line 122, please remove the sentence, “Further clinical and radiographical follow-ups should be performed to assess the stability of the treatment over time.” This sentence is not appropriate for the case report section.
4. In this case, ankyloses of the replanted tooth was noted. In the following guideline,
Andersson L, et al. International Association of Dental Traumatology guidelines for the management of traumatic dental injuries: 2. Avulsion of permanent teeth. Dent Traumatol. 2012;28:88-96. doi: 10.1111/j.1600-9657.2012.01125.x.
Ankylosis can be highly anticipated in the case of replanted immature teeth with dry time longer than 60 min. Once, ankylosis is present, decoronation may be necessary according the severity of infraposition. Please develop this idea in the discussion section.
5. In the discussion section (especially Page 7, Line 157), the paragraph seems not to be appropriate. The technique in this case report is not regenerative endodontic procedures but apexification. Thus, this paragraph should be revised.
Author Response
REVIEWER 2
This case reports demonstrates a treatment strategy regarding a necrotic immature permanent tooth with delayed replantation. The topic is of clinical significance in the field of pediatric dentistry. However, there remained some concerns.
The Authors are grateful for the Reviewer’s considerations. The case report has been edited as recommended to be eligible for publication.
- I recommend to revise tooth numbering system for readability. Please consider to provide FDI-two digit tooth numbering system or the full name of the teeth.
Tooth numbering has been revised and provided according to FDI Numbering System as suggested.
- Please provide a PRICE 2020 checklist (table 1) as supplementary files, instead of table.
Table 1 has been removed from the text and presented as supplementary material as suggested.
- In the Page 6, Line 122, please remove the sentence, “Further clinical and radiographical follow-ups should be performed to assess the stability of the treatment over time.” This sentence is not appropriate for the case report section.
The sentence has been deleted as recommended.
- In this case, ankyloses of the replanted tooth was noted. In the following guideline, Andersson L, et al. International Association of Dental Traumatology guidelines for the management of traumatic dental injuries: 2. Avulsion of permanent teeth. Dent Traumatol. 2012;28:88-96. doi: 10.1111/j.1600-9657.2012.01125.x. Ankylosis can be highly anticipated in the case of replanted immature teeth with dry time longer than 60 min. Once, ankylosis is present, decoronation may be necessary according the severity of infraposition. Please develop this idea in the discussion section.
The suggested points have been added and developed within the Discussion section according to the Reviewer’s recommendation.
- In the discussion section (especially Page 7, Line 157), the paragraph seems not to be appropriate. The technique in this case report is not regenerative endodontic procedures but apexification. Thus, this paragraph should be revised.
The Authors agree with the Reviewer’s concern and have wholly revised the paragraph to avoid misunderstanding. The regenerative endodontic procedures have been quoted only to provide an alternative in the management of traumatized teeth, as reported in the scientific literature.
Round 2
Reviewer 1 Report
The manuscript improved from the first version.
However, authors changed wording at conclusions, as suggested, but forget to adjust the abstract. Therefore, I suggest to replace the word “contrast” by “restrict”.
Reviewer 2 Report
I am satisfied with the revised manuscript.
Thank for the efforts.